# Fluorescence Microscopy in Adeno-Associated Virus Research

**DOI:** 10.3390/v15051174

**Published:** 2023-05-16

**Authors:** Susanne K. Golm, Wolfgang Hübner, Kristian M. Müller

**Affiliations:** 1Cellular and Molecular Biotechnology, Faculty of Technology, Bielefeld University, 33615 Bielefeld, Germany; susanne.golm@uni-bielefeld.de; 2Biomolecular Photonics, Faculty of Physics, Bielefeld University, 33615 Bielefeld, Germany; whuebner@physik.uni-bielefeld.de

**Keywords:** microscopy, adeno-associated virus, AAV, AAV labeling, DNA labeling, laser scanning confocal microscopy (LSCM)

## Abstract

Research on adeno-associated virus (AAV) and its recombinant vectors as well as on fluorescence microscopy imaging is rapidly progressing driven by clinical applications and new technologies, respectively. The topics converge, since high and super-resolution microscopes facilitate the study of spatial and temporal aspects of cellular virus biology. Labeling methods also evolve and diversify. We review these interdisciplinary developments and provide information on the technologies used and the biological knowledge gained. The emphasis lies on the visualization of AAV proteins by chemical fluorophores, protein fusions and antibodies as well as on methods for the detection of adeno-associated viral DNA. We add a short overview of fluorescent microscope techniques and their advantages and challenges in detecting AAV.

## 1. Introduction

There are several reviews covering adeno-associated virus (AAV) biology and a plethora of reviews covering light microscopy techniques but almost none combining these two aspects. We feel that the importance of the intersection of these two research areas justifies its own review article. Hereby, we take an AAV-centric view. To keep the focus, we do not include work which primarily uses AAV as a transfection agent.

### 1.1. AAV Genes and Structure

The non-enveloped AAV belongs to the *Parvoviridae* family and was first described in 1965 [1]. Many different serotypes of AAV have been discovered, whereby stereotype 2 (AAV2) is historically the best studied [2,3]. The AAV consists of a 4.7 kb long single-stranded DNA genome encapsidated in an icosahedral capsid with a diameter of about 25 nm, a T = 1 symmetry, and with 2-, 3-, and 5-fold symmetry axes. With its protruding loops at the 3-fold axis and a hole at the 5-fold axis, the capsid resembles a deltoidal hexecontahedron. The AAV capsid consists of three capsid (Cap) proteins named VP1, VP2, and VP3 which form the capsid with 60 subunits in a ratio of about 1:1:10 (VP1:VP2:VP3) [4,5]. The viral DNA is flanked by two inverted terminal repeats (ITRs) consisting each of a 145 bp long mostly self-complementary sequence, which forms a T-shaped hairpin structure [6]. The viral genome encodes, among others, the non-structural Rep proteins Rep78, Rep68, Rep52, and Rep40 as well as the structural Cap proteins VP1, VP2, and VP3. The transcription of viral proteins is initiated with the help of the three promoters p5, p19 and p40. Transcription of the larger non-structural Rep proteins Rep78 and Rep68 is initiated by the p5 promoter and of the shorter Rep52 and Rep40 by the p19 promoter, whereby alternative splicing in each case yields the smaller variant. Expression of the Cap proteins is mediated by the p40 promoter. The different VP proteins are produced by alternative splicing and a leaky scanning mechanism [7,8]. In addition, a region in the *cap vp1* gene encodes the non-structural membrane-associated AAV protein (MAAP), which is a potential AAV egress factor [9,10,11]. A second region overlapping all the cap genes encodes the assembly activating protein (AAP). Transcription of the AAP mRNA is initiated by a non-canonical start codon (CTG) [12]. Another protein, called protein X, is also encoded by the cap region. The functions of protein X are not known yet; however, it might play a role in DNA replication and enhance vector DNA replication [12,13].

### 1.2. AAV Life Cycle

Replication of AAV depends on coinfection with a helper virus such as the adenovirus (Ad) [14]. In the absence of a helper virus, the AAV enters a latent cycle and integrates preferably into the AAV site on chromosome 19 [15]. Whenever a suitable helper virus is present, the lytic cycle is initiated.

In the case of adenovirus, helper functions of proteins E1a, E1b, E4 orf6, DBP and VA RNA have been identified. They exert many effects, such as influencing AAV transcription (e.g., activation of the p5 promoter) and second strand synthesis [16,17,18]. The lytic life cycle of AAV is complex and comprises several steps, from infection of the host cell via the expression of viral proteins, replication of the viral DNA, capsid formation, and DNA packaging to yield the final AAV [19].

For cell entry, the virus first binds to a primary receptor which promotes interaction with a secondary receptor. For serotype 2, the primary receptor is heparan sulfate proteoglycan (HSPG) [20,21,22]. A further receptor found to be essential for most serotypes is the AAV receptor (AAVR), which is a type 1 transmembrane protein encoded by the KIAA0319L gene [23].

For the endocytosis of AAV2, several co-receptors were identified, such as the hepatocyte growth factor receptor (HGFR) [24], laminin receptor (LamR) [25], or integrins (α5β1 and αvβ5) [20,26]. Binding of the secondary receptor normally triggers virus internalization by clathrin-mediated endocytosis in dependence on dynamin proteins [27,28]. Nonnenmacher et al. described the pleomorphic clathrin-independent carrier (CLIC)/GPI-enriched endocytic compartment (GEEC) pathway as the major endocytic infection route for AAV2 [29].

The intracellular transport paths of AAVs have been studied extensively but remain incompletely resolved [30]. AAV2 is known to use endosomal retrograde transport after cell entry followed by endosomal escape. It has been shown that acidification of the endosome induces structural changes such as exposure of the unique VP1 domain that mediates the escape [31,32]. For the latter, the VP1 protein contains in its unique N-terminal part of a phospholipase A_2_ (PLA) domain that cleaves off acyl-esters in position two of phospholipids, which weakens the membrane [33,34,35]. In addition, the N-terminal domain harbors three basic regions (BR1–BR3) with BR2 and BR3 also being present on VP2, which from a basic cluster that functions as nuclear localization sequences (NLS). Two more basic regions (BR4, BR5) are part of all three VP proteins [27,36,37].

Another point of interest is the interaction of the viral capsid with entry factors in the endocytic and trans-Golgi compartments. A recently discovered, highly conserved entry factor is GPR108, which may play an important role in endosomal escape and nuclear import [38]. This new discovery challenges a previous model describing the endosomal escape into the cytoplasm, which is followed by transport across the cytoskeletal network into the perinuclear region [28,39,40]. Next, the viral DNA enters the cell nucleus, and the viral genome is uncoated. Yet again, the driving forces of these processes are not fully resolved. A review by Mattola et al. covering parvovirus–nucleus interactions provides a broader scope on this topic [41].

In the lytic cycle, synthesis of the second DNA strand takes place, and Rep and Cap proteins are expressed [42]. In the following, the capsids are assembled and then loaded with ssDNA. Since capsid assembly takes place in the nucleus and herein specifically in the nucleolus [12,43], the VPs have to be post-translationally transported there. For this purpose, VP proteins contain the above-mentioned nuclear localization signal (NLS) [36,44,45]. VP3 by itself is present in the nucleus but remains excluded from the nucleolus. Nucleolar localization and capsid assembly is mediated by AAP [12]. However, it has also been shown that the contribution of AAP to capsid assembly is gradual to varying degrees depending on the serotype [46,47]. Rep proteins also contain an NLS and can enter the nucleus via importin-α [48]. The viral genome ssDNA is produced by strand displacement mediated by Rep78 and Rep68, and the resulting + and - strands are packed into the capsid in the same ratio. The ITRs serve as packaging signals and are recognized by the Rep proteins Rep52 and Rep40. They bind to the empty capsid and, with the help of their helicase activity, the ssDNA is passed through the cylindrical capsid pore of the 5-fold axis into the capsid [49].

## 2. Fluorophores and Labeling

The unique fluorescence and target specific localization of a fluorophore are the desired features for observing the protein of interest (P.O.I.). Fluorophores can be of synthetic–chemical nature or genetically encoded as a fluorescent protein (FP). Figure 1 shows the basic concepts of protein labeling. Fluorescence microscopy is a versatile technique. It allows for an overview assessment as well as the analysis of specific questions with high time and spatial resolution and options for relative quantification supplementing or superseding classical methods such as Western blot. In the field of AAV research and vector development, it enables the clarification of fundamental questions, such as the cellular distribution of AAV proteins, and of more specific interactions and colocalization, e.g., with host cell proteins. Microscopic investigations have made major contributions to the understanding of AAV biology literally illuminating the viral life cycle, virus–cell interaction, endocytic pathway, disassembly, nuclear processing and virus assembly. For this, fluorescent labels have been applied for the visualization of virus particles as well as for individual components such as VPs or the viral DNA genome. The microscopy techniques mentioned in this review are listed in Appendix A.

The choice of fluorophores alone can be critical with brightness, photostability and the spectral match to the equipment being important. Chemical fluorophores used to have advantages in comparison to fluorescent proteins with higher extinction coefficients, higher quantum yield, better photostability and narrower excitation and emission spectra. However, some chemical fluorophores as well as applications utilizing antibodies are often unsuited for live imaging, because these are either not membrane permeable or can impair the biological function of the P.O.I. Fluorescent protein technology became better in all aspects, often matching or surpassing chemical fluorophores, but additional factors such as maturation, turnover, and multimer status should be considered. In general, the individual photon budget of the fluorophores before bleaching as well as the possible cytotoxic effect due to the fluorescent process are critical.

### 2.1. Labeling with Amino Acid-Reactive Dyes

The covalent labeling of AAVs with fluorophores has been applied in many ways. A widely used family of covalent fluorophores is derived from cyanine dyes. They are built up of two indole groups which are connected by a polymethine bridge. The dyes of this family differ in the length of the polymethine bridge, modifications of the heterocycle, which determines their optical properties, and by the added coupling chemistry. The advantages of these dyes are their high brightness and photostability. Bartlett et al. linked amine reactive Cy3 or Cy2 covalently to AAV2 or AAV3 capsids to monitor their neuron specific uptake in infused brain samples or their distribution in HeLa cells [50,51,52]. In 2001, Seisenberger et al. used Cy5 covalently linked to particles to study the entry pathway of single AAV viruses into living cells by recording individual trajectories in real time. A Gaussian fit strategy yielded about 40 nm resolution in image series recorded in 40 ms intervals, which pioneered super-resolution microscopy in AAV research [53]. Collectively, these observations revealed kinetics of the docking events of the viruses to HeLa cell, their endocytosis, as well as their transport through the cytosol into the nucleus by means of motor proteins and nuclear tubular structures. Dalkara et al. labeled AAV 1, 2, 5, 8, and 9 with amine reactive Cy3 to establish expected particle localization upon uptake in HEK-293T cells, and they used co-staining with anti-AAV1, 2, and 5 antibodies to ensure biocompatibility of the label. Labeled particles were then intravitreally administered in rats, and their accumulation at the vitreoretinal junction was imaged by fluorescence microscopy. AAV 2, 8, and 9 bound but only AAV 2 transduced inner cells, while AAV1 and 5 did not attach [54]. Nicolson et al. used Cy5-labeled rAAV2 to show that rAAV2 enters the nucleus via the nuclear pore complex and interacts with importin-β [44].

In the 2020 work of Mével et al., the aim was to achieve a higher transduction efficiency. Therefore, amine coupling via the traditional isothiocyanate reaction was used to attach ligands to the capsid, which was first evaluated with fluorescein isothiocyanate (FITC) [55].

The development of derivatives of cyanine and fluoresceine as well as from other basic fluorophores led to the availability of many dyes with more specific fluorescent spectra spanning a wide range of wavelengths while providing varying pH stabilities, varying Stokes shifts, greater photostability and, last but not least, higher extinction coefficients and quantum yields, which as product are called brightness. Manufacturers tend to brand their fluorophores under one umbrella name (e.g., AlexaFluor, Atto) but neither the umbrella name from one company nor the name-determining wavelength close to the absorption peak maximum between alternative dye series necessarily indicates similar underlying fluorophores. In addition to the organic chemistry-based fluorophores, quantum dots have been established, which are nanometer-sized semiconductors coated with organic molecules for biocompatibility and attachment. General protocols for the genetic modification of surface-exposed positions and the chemical labeling of AAV have been published [56].

In the context of AAV, AlexaFluor 488 was used for covalent labeling of AAV capsids and observed during the transduction of HeLa cells. Xiao et al. used AlexaFluor 488-conjugated AAV2 wild type besides other techniques (A20 monoclonal antibody immunocytochemistry and subcellular fractionation techniques followed by DNA hybridization) to analyze the dependency of AAV on adenovirus coinfection. In the presence of adenovirus, nuclear translocation was facilitated, whereas endosomal escape seemed not to be influenced [57]. Joo et al. compared conventional AlexaFluor-labeled AAV2 with their amine reactive method of labeling AAV2 with quantum dots in long-term live-cell imaging. In this study, they tracked single particles and showed that the viruses infected the cells via the clathrin-dependent pathway made their way through endosomes, exploited the cytoskeleton network for transport through the cell and involved the ubiquitin–proteasome system in nuclear transport [58]. Furthermore, AlexaFluor 568-labeled AAV2 were used to study the colocalization with AAVR in HeLa cells [59].

### 2.2. Labeling with Click Chemistry

Another interesting method for covalent labeling is the so-called click chemistry, which allows for the covalent coupling of different fluorophores typically via Copper(I)-catalyzed azide-alkyne cycloaddition (CuAAC) or strain-promoted azide-alkyne cycloaddition (SPAAC). To introduce the unique chemical handle in the P.O.I., expansion of the genetic code with unnatural amino acids is used. This method has already been used in AAV studies. For example, Zhang et al. used in vitro click chemistry-based fluorescently labeled AAV capsids to study transduction in HeLa cells. For this, an azide-containing unnatural amino acid (UAA, Nε-2-azideoethyloxycarbonyl-l-lysine) was genetically introduced into the AAV2 capsid. The residue site (R447) was chosen to not affect viral assembly or infectivity. By click chemistry, DIBO-tagged fluorophore ligands were conjugated to the modified capsid. In combination with colocalization studies with Rab5, Rab7 and Rab11, they showed that early, late and recycling endosomes are involved in the successful transduction of AAV2. Furthermore, they confirmed the role of actin filaments in the intracellular transport and nuclear entry via co-staining of actin filaments and treatment with Cytochalasin B, which disturbs microfilaments [60]. Katrekar et al. targeted and analyzed various amino-acid positions for UAA incorporation and identified S578 or N587 as preferable sites. They coupled DIBO-AlexaFluor 594 to the surface of AAV2 and expanded the work by coupling oligonucleotides or lipids to AAV-DJ. The virus crosslinked with DBIO-AlexaFluor was added to HEK-293T cells and imaged 2 h after treatment [61].

### 2.3. Labeling via Tags

Chemical fluorophores can be covalently attached to the P.O.I. also with diverse systems of genetically introduced affinity tags combined with specific enzymes expressed in a live cell environment. AAV9 have been labeled through the insertion of a 12 aa tetracysteine motif in the viral capsid and the coupling of biarsenic or maleimide modified dyes. Such modified AAV9 particles were injected intravenously in mice followed by time-lapse intravital microscopy. Imaging of the brain 2 h post injection showed accumulation juxtapositioned around the nuclei. The group also used chemical biotin coupling to the tag for pull-down experiments [62]. Another approach to biotinylation was the insertion of a 15 amino acids long biotin acceptor peptide (BAP) in the capsids. The biotin isostere ‘ketone-1’ with the ureido nitrogens replaced by methylene groups was conjugated to AAV1 catalyzed by the *Escherichia coli* biotin ligase (BirA) followed by chemical coupling of AlexaFluor 488 hydrazide to the ketone. In this way, particle binding and intracellular particle distribution in human umbilical vein endothelial cells was observed [63].

Another tag, the FLAG-tag, was used in 1999 to detected VP3 in Western blots [64]. The FLAG-tag consists of the eight hydrophilic amino-acids DYKDDDDK and is detected by commercially available antibodies [65]. A FLAG-VP2 fusion was used for AAV immobilization and subsequent detection in CHO cells [66]. Earley et al. N-terminally fused the FLAG-tag to AAP at the genetic level to different AAV2 mutants to study nuclear and nucleolar localization signals of AAP [67]. They also used this technique to investigate subcellular localization and assembly-promoting effects of AAP of serotypes 1 to 12 for homologous as well as heterologous VP3 proteins by fluorescence microscopy. In this study, FLAG-tagged AAP variants were used, and the results showed that AAV 4, 5, and 11 assembled without requiring AAP [68].

For the visualization of the recently discovered MAAP, Elmore et al. used tagging of the protein at the genetic level to enable the use of readily available antibodies. They fused the HA-tag (amino acid sequence YPYDVPDYA) C-terminally and incubated fixed cells with a primary rabbit polyclonal anti-HA antibody and then a Alexa 647-labeled goat anti-rabbit antibody [10].

### 2.4. Labeling with AAV Protein-Binding Antibodies

Antibodies are an important tool for the highly specific labeling and identification of proteins of interest and are extensively used in fluorescence microscopy and AAV research. Antibodies enable labeling without any structural interference in the first place. However, this method requires fixation and permeabilization of the cells to provide access for the antibodies. Therefore, this technique is not suited for live-cell imaging but enables a close look at specific time points and yields fundamental information such as the localization and colocalization of P.O.I.s. In Table 1, we list antibodies with different binding modes toward AAV proteins. For microscopy or ELISA, antibodies typically recognize exposed epitopes, which could be linear, thus being compatible with Western blot, but also discontinuous when formed by the folded protein or protein complex. The primary, target specific antibodies can be directly fluorescently labeled. However, using a secondary labeled antibody provides experimental flexibility while being cost effective. Monoclonal antibodies are advantageous, because of their defined properties and reproducibility, yet easy to obtain polyclonal sera are still used.

In AAV research, the mouse monoclonal antibody (mAb) A20 has been established in early work in the group of Kleinschmidt along with anti-VP mAbs A69, B1 and anti-Rep mAbs 76/3 (also named 76.3) and 303/9 [80]. A20 is the prototype antibody only recognizing assembled capsids of AAV serotype 2. The same group demonstrated the use of these antibodies in fluorescence microscopy to track subcellular location during transduction and AAV formation in combination with DNA in situ hybridization [43]. The binding modes of the A20 antibody as well as the newly established conformational antibodies C24-B, C37-B, and D3 were characterized by Wobus et al. in 2000, who also identified the linear epitope sequences of the A1, A69 and B1 antibodies. The mAb D3 recognizes serotypes 1 to 9. Antibody A1 recognizes the VP1 specific N-terminal peptide sequence KRVLEPLGL. The VP1 and VP2 proteins are bound by antibody A69 via aa 171 to 182 (VP1 numbering). Antibody B1 recognizes all three VP proteins (VP1, VP2 and VP3) and has pan-serotype specificity except for AAV4 via the common C-terminal amino acids. These amino acids are located inside the capsid, and thus, B1 detects free VP protein [69]. The group of Kleinschmidt later introduced the intact capsid antibodies ADK1a, ADK1b, ADK4, ADK5a and ADK5b [81] as well as ADK6, ADK8, and ADK9 [46].

As described in several citations in Table 1, a detailed picture of the binding modes of these antibodies including contact and footprint residues was obtained by cryo-electron microscopy and image reconstruction of complexes comprising the antibodies bound to capsids, which is a technique often published by the group of Agbandje-McKenna [70,74,76,77,78]. Most of the listed antibodies are commercially available from several vendors.

Over the years, especially A20 aided in many microscopy studies analyzing AAV transduction and production. For example, the colocalization of AAV2 with nucleolin confirmed the interaction [82]. Johnson et al. used this antibody to localize full and empty AAV capsids relative to nucleolin with confocal microscopy and z-stacks as well as the effect of hydroxyurea or the proteasome inhibitor MG132 on AAV2 localization in HeLa cells [83]. Both agents improved transduction, and these studies were expanded and combined with AAV mutants with altered basic regions (BR). In addition, the antibody A1 was used in this study, which binds to the N-terminus of the AAV2 VP1 protein. Effects on localization and transduction were followed by microscopy, and one mutant localized to the nucleus but did not induce the expression of their delivered gene of interest [84]. The transduction of monocyte-derived immature dendritic cells by AAV2 was observed with mAb20, Lamin B1 and DAPI staining [85].

The group of Asokan used the A20 mAb to study the chemical modulation of endocytotic sorting of AAV. For their confocal imaging, they also used immunofluorescence staining of valosin-containing protein (VCP), which interacted with the transduction-enhancing compound eeyarestatin I (EerI) as well as the organelle markers EEA1 for early endosomes, Rab7 for late endosomes, LAMP1 for lysosomes, and Golgin-97 for Golgi [86]. In a further study, the group used the A20 mAB in combination with ring finger protein 121 (RNF121) staining, which is a protein identified as a regulator of AAV transcription [87]. They also identified the Golgi compartment-resident ATP-powered calcium pump (secretory pathway calcium ATPase 1 (SPCA1)) encoded by the *ATP2C1* gene as a critical player for AAV sorting. The confocal microscopy of control and ATP2C1 knockout Huh7 cells with mAb A20 or mAb A69 and anti-golgin-97 staining demonstrated altered trafficking [88].

The A20 [89] and ADK5a [90] antibodies have also been used to study AAV particles produced in *E. coli.* Interestingly, a single-chain Fv (scFv) A20 Fc antibody construct displayed slightly less stringent structural requirements for binding [89].

The microscopic detection of Rep proteins was already performed in 1992 by Hunter et al. for imaging the (co-)localization of AAV Rep and Cap proteins in the nucleus of infected cells. They used, among others, mono-and polyclonal Anti Rep78/68 and Anti Rep 52/40 antibodies, combined with rhodamine-conjugated goat anti-rabbit or fluorescein-conjugated goat anti-mouse secondary antibodies [91]. Images were recorded on film from a widefield fluorescence microscope. The use of recording methods predating sensitive digital cameras required long exposure times, yet the quality of the images is remarkable. Since then, anti-Rep antibodies have been used in several immunofluorescence studies. The group of Weitzman studied the nuclear import of Rep proteins and various mutants thereof, especially of the NLS sequences, with antibodies and GFP fusions [48]. They also determined the colocalization and requirement of HSV-1 helper proteins and the interaction of Rep proteins with single-stranded DNA [92]. Heilbronn et al. and Slanina et al. determined colocalization of the Rep proteins with AAV ssDNA and with ICP8 at the nuclear HSV replication sites [93,94]. Polyclonal rabbit sera were used to establish the colocalization of AAV Rep and Cap proteins and B23/Nucleophosmin (NPM) [95].

The VP3 capsid assembly of various serotypes with original or AAV2-derived AAP was detected with a polyclonal serum for the VPs and the capsid-specific antibodies mAb A20, ADK1, ADK4, ADK5a, ADK8, or ADK9 [46]. Earley et al. used, besides the FLAG-tagged AAPs, the mABs A20, B1, ADK5, ADK8, ADK8/9 and a polyclonal Ab for nucleostemin for colocalization studies [67,68].

Galibert et al. detected MAAP with a custom generated polyclonal antibody generated toward a MAAP peptide (MAAP-GALKKI antibody) followed by an anti-rabbit AlexaFluor-conjugated secondary antibody [11].

### 2.5. Labeling with Genetically-Fused Fluorescent Proteins

Fluorescent proteins (FP) became a cornerstone of microscopy since the introduction of the Green Fluorescent Protein (GFP) in 1994 [96]. Since then, fluorescent proteins are designed based on sequences from species such as the jellyfish Aequorea Victoria and other diverse cnidaria to cover the entire visible light spectrum and provide a high brightness and good photostability [97]. In addition, variants with different pKa values, chromophore maturation times and Stokes shifts have been developed. For specific applications, photoactivatable or photoswitchable variants are available. FPbase (https://www.fpbase.org/; accessed on 1 March 2023) provides a good overview of the ever-growing options [98]. The DNA encoding a given fluorescent protein is typically fused to the DNA encoding the protein of interest or alternatively to a specific tag system such as targeting sequences or binding domains. The FP can be inserted at the amino terminal or carboxy terminal end as well as internally. Although the FP coding sequences are from diverse species or are synthetically designed, the fluorescent proteins fold to similar beta-barrel structures with a molecular weight of around 27 kDa. Some FP proteins are multimeric, while some monomeric variants retain a tendency to multimerize at higher concentrations. Even a monomer can impact the fused P.O.I. with regard to folding, maturation, localization, trafficking and function. Nevertheless, FP fusion is a very effective tool and enables a better understanding of AAV biology.

Fusion proteins are commonly used to study the localization of AAV proteins. In 2004, Cassell et al. used GFP-tagged Rep68 and Rep78 proteins for the detection of a C-terminal NLS in these Rep proteins [48]. Fraefel et al. fused the first 522 codons of *rep* with in two variations with the DsRed2 or the eCFP coding sequence [99,100].

Lux et al. used VP2-GFP fusion proteins for the visualization of viral trafficking in transfected HeLa cells [101]. Another fusion protein was developed by Judd et al., who successfully introduced the fluorescent protein mCherry into the VP3 domain, allowing the production of infectious AAVs [102].

In addition to the FLAG-tag, Earley et al. made different fusion proteins with AAP segments and a C-terminal GFP to study nuclear and nucleolar localization signals in AAP of AAV2 [67]. Elmore et al. showed recently that the MAAP is an AAV egress factor. In their study, MAAP was modified with eGFP to analyze its function for different AAV serotypes by the confocal imaging of AAV-producing HEK-293 cells [10].

### 2.6. Staining of DNA and RNA

Another important element for understanding AAV biology is the viral DNA, which can also be examined microscopically using various techniques. Figure 2 provides an overview of the DNA staining techniques used in AAV research, which are mentioned below.

In 1996, Weitzman et al. visualized AAV wild-type DNA in HeLa cells infected with AAV and adenovirus. For the visualization of DNA, they used in situ hybridization and a confocal scanning microscope. These results underlined the dependency on adeno viral coinfection for AAV replication by the detection of AAV genomes in replication centers [103].

Another method for the visualization of active DNA synthesis was used by Stracker et al. in 2004. They were interested in DNA production during AAV replication. For this purpose, they transfected Vero cells, incubated the cells with bromodeoxyuridine (BrdU) after 16 h, fixed the cells and used an anti-BrdU antibody to detect the labeled DNA (Figure 2A). Their studies showed that AAV DNA replication takes place in nuclear compartments, where AAV Rep proteins are present. These nuclear compartments increased over time. Furthermore, they observed a positive impact of single-stranded DNA-binding protein of Ad, HSV-1 (ICP8) and cellular replication protein (RPA) on AAV replication by colocalization studies [92].

In addition to BrdU and as an in vivo DNA painting approach, Fraefel and colleagues used repeated insertions of a repressor target site in a recombinant AAV genome and concomitant expression of the repressor protein fused to a fluorescent protein (Figure 2B). The repressor binds the target site in the living cell, leading to a signal distinct from the background. For this purpose, 40 lac repressor-binding sites (lac operator, lacO) were inserted between ITRs, and the lac repressor/inductor (LacI) fused to eYFP was provided by a second plasmid [99]. This yielded a time course of dsDNA formation and localization, which overlapped with Rep but not promyelocytic leukemia (PML) nuclear bodies (NBs). This approach was extended to the Tet repressor (TetR) and the tet operator (tetO). The Tet repressor was fused with either eYFP or eCFP, and 35 copies of tetO were used as a target. This enabled the simultaneous detection of ITR and p5 promoter-bearing plasmids in the context of HSV-1 [100]. The LacI-eYFP/lacO system was used by the group of Weitzman to detect proximity of the rAAV foci with foci of the DNA damage response [104]. Furthermore, they used the system in combination with other methods to demonstrate that degradation of the MRN (Mre11/Rad50/Nbs1) complex, but not p53, by adenoviral E1b55K/E4orf6 enhances AAV dsDNA accumulation upon transduction and thus provides an essential helper function. In the same publication, they also provide fluorescence microscopy images stained for Nbs1, DBP, Mre11, Rad50 and Rep [105].

Branched DNA signal amplification was introduced for virus detection in the early 1990s [106]. This technology concept was continuously developed, and recently, modern variants have been used to visualize AAV DNA (Figure 2A). The ‘signal amplification by exchange reaction fluorescence in situ hybridization’ (SABER-FISH) method is more sensitive than the standard FISH method. This technique endows oligonucleotide-based FISH probes via primer exchange reaction (PER) cycling with long, single-stranded DNA concatemers that hybridize with multiple short complementary fluorescent imager oligonucleotides [107]. It provides up to multifold signal amplification and can thus detect smaller DNA fragments, which is of great advantage in the context of the small AAV genome. In 2020, Wang et al. used this technique to detect in situ AAV8 vector genomes in tissue injected with the virus. Using different colors for the 5′-end and the 3′-end of the rAAV DNA in combination with colocalization enabled them to judge genome integrity. For visualization, they used a laser-scanning confocal microscope [108].

For enhanced specificity of the nucleotide branch strategy, the RNAscope technology uses two target specific ‘z’ probes which are bridged by one pre-amplifier oligonucleotide (Figure 2A) [109]. This was used to detect AAV DNA and RNA. AAVs were injected into the tissue, and the viral DNA or RNA was detected in situ at different time points in various tissues using bright field microscopy [110].

In 2022, Sutter et al. analyzed AAV2 uncoating in transduced NHF cells by combining the FISH technique for AAV2 DNA with the immunofluorescence of assembled capsids, the proteins VP1 and VP2, as well as the cellular proteins fibrillarin and cyclin A. The analysis of confocal microscopy suggested that the uncoating of AAV2 is a stepwise process that is influenced by the cell cycle and involves changes in the nucleolar structure [111].

Recently, Jang et al. deployed a new technique called ultrasensitive, sequential fluorescence in situ hybridization (USeqFISH) for spatial transcriptomics in fixed cells. This method uses a padlock paired with a primer for detection (called SNAIL logic for “specific amplification of nucleic acids via intramolecular ligation”) [112] and amplifies the signal by the combination of rolling circle amplification (RCA) [113] followed by a hybridization chain reaction (HCR) [114] collectively termed RCAHCR [115]. The signal amplification is decoupled from the target detection and is based on unique identifiers in the padlocks, which enabled quenching (i.e., stripping) of the HCR product with a toehold oligonucleotide for sequential detection. They showed that their technique detected RNA with, e.g., four probes with an increased mean signal intensity compared to that of HCR or RCA methods. In this way, the detected number of endogenous and viral genes was increased. By using USeqFISH, they studied six AAV variants packaging VP3 genes in context to transduction and cell subtype tropism in the mouse brain [115].

## 3. Microscopy Methods

In the following section, we give an overview of fluorescent microscopy methods applicable to the understanding of cellular and viral mechanisms of AAV (Figure 3A). Because there is hardly a single universal method available, we mention the individual strengths and disadvantages. Microscopy requires the synergy of contrast, magnification and resolution. While contrast is achieved through specific labeling combined with bright fluorescence, the light intensity emitted by the sample is recorded after being convolved through the optical system. Thus, microscope techniques and hardware determine magnification and associated resolution. Generally, fluorescence microscopy techniques are subject to a physical absolute diffraction determined resolution limit. Ernst Abbe described in 1873 this relation between the wavelength of light and the numerical aperture of the lens as well as its immersion medium refraction index used for excitation and emission. In practice, with the highest magnification lenses available, this leads to resolution limits in the visible spectrum around 200 nm in lateral and 500 nm in axial dimensions.

### 3.1. Widefield Microscopy

Historically, the most accessible fluorescent microscopy method is widefield epifluorescence microscopy (Figure 3B). In addition to being low-cost, this imaging method has relatively fast temporal and good spatial resolution [117]. In widefield illumination mode, light passes through the entire sample in the z-dimension inducing fluorophore emission above and under the actual focal plane. As a consequence, the background contributes to recorded intensities. The reduction in this background is the key to a significant increase in image quality. The obtained image is a diffraction limited result of the convolution of the real object intensities by the optical system. The 3-dimensional distribution of small light sources recorded as a point-spread function defines the system. Various deconvolution algorithms, which take into account the 3-dimensional distribution of a light source in the system, can minimize the out-of-focus background and increase the in-focus signal contrast.

### 3.2. Confocal Microscopy

True optical sectioning is achieved digitally structural illumination modes (Figure 3B). The development of laser scanning confocal (LSCM) microscopes leads to a broad popularization of the use of fluorescent microscopy. Unlike widefield fluorescence microscopy, the excitation light is guided to form a single diffraction limited spot. From that point, the emitted light is spatially filtered by a physical or digital pinhole, therefore blocking the out-of-focus light [118]. Depending on the machines, illumination is performed with either a single point or several points at once. LSCMs are versatile commercially available instruments. Nevertheless, they have the disadvantages of higher associated costs and possibly very slow image acquisition times. The latest developments, especially in the integration of fast resonance scanners, can mitigate the recording speed issue for live imaging. Multi-spot confocal microscopy, such as for example spinning disc laser confocal microscopes, can make very fast imaging possible, albeit usually at the cost of resolution. In addition to the obvious volumetric multi-color imaging capacities, the illumination mode can be applied to spatial–temporal analysis of fluorescent samples. FRAP (fluorescence recovery after photobleaching) and FLIP (fluorescence loss in photobleaching) measurements are rapidly applicable methods in confocal microscopes to measure the global diffusion rates of fluorescently labeled molecules [119]. FRAP experiments in combination with fluorescence fluctuation microscopy and with the use of a photoactivatable fluorescent-protein fusion to capsid proteins were used to study the intranuclear dynamics of canine parvovirus and of host cell components [120,121].

Interactions and structural changes can be observed very precisely with FRET (Förster resonance energy transfer) or FCS (fluorescence correlation spectroscopy)/FCCS (fluorescence cross-correlation spectroscopy) [122]. While FRET can readily be applied in living cells to sense interactions, it is more difficult with FCS or FCCS, which rely on precise volume measurements. Cellular movements and subtle changes in the orientation of the observed compartments impact diffusion-rate measurements, thus requiring technical adaption (e.g., fast measurements aided by line scanning) and biological controls. These techniques could contribute to the understanding of molecular interactions and structural changes as well as overall dynamics of AAV molecules components during its life cycle. Image correlation methods including auto-, pair- and cross-correlation, and number and brightness analysis were used to study the entry pathway of AlexaFluor 594 or 488-labeled canine parvovirus at the single-particle level in living feline kidney cells expressing lamin C-EGFP or importin β-GFP [123].

### 3.3. Total Internal Reflection Fluorescence Microscopy

Another illumination mode eliminating out-of-focus contributions is the total internal reflection fluorescence microscopy (TIRFM) (Figure 3B). An excitation laser beam is reflected from the cover glass carrying the sample with a significant difference in refractive index leading to an evanescence wave propagating along the surface of the glass. This restricts the excitation to about 150 nm in depth above the surface of the glass [124]. This method is limited to image contact areas between the sample and the glass it is placed on. Variations of this method create a near/close to total internal reflection light sheet going further away from the cover glass into the sample [125]. Typically, TIRFM is associated to widefield illumination microscopes, giving the advantages of possible very fast acquisition even if only restricted to one focal plane.

### 3.4. Super Resolution Microscopy

Several methods to resolve fluorescent objects up to single molecules under the diffraction limit have been developed over the years and defined as super-resolution methods (reviewed in [116]) (Figure 3C).

These methods can be described for example in three major technical categories. Reversable Saturable Optical Fluorescence Transitions (RESOLFT) is in principle reducing the fluorescence process, therefore shrinking the emitted signal possibly in all 3 dimensions. Here, the concept is most known from well-established implementations such as STimulated Emission Depletion (STED) or Ground State Depletion (GSD) [126].

Another super-resolution technique is the Single Molecule Localization Microscopy (SMLM). Here, fluorophores are manipulated for example through stochastic switching/blinking to become sparse in their timely emission. Blinking molecules emit at a given time point as a single molecule with no other emitter in its resolution-limited surrounding space. The precise localization of the molecule is determined through a simple signal fitting procedure. Finally, the multiple localizations of single molecules are used to render a super-resolved image. Variations of this technique include the ability, for example, to determine an axial localization based on purposely introduced aberrations or to calculate higher resolutions based on fluctuations of more than one molecule in a diffraction limited spot.

Lastly, the third category can be defined as Super-Resolution Structured Illumination Microscopy (SR-SIM). This principle relies on the physical properties of frequencies mixing, where the application of a known illumination structure, such as a sinusoidal pattern, brings resolution information from beyond the diffraction limit into the limited image. Multiple images with phase and angle shifts of the pattern are necessary to cover the entire sample. Algorithms identify these signals and recombine them into a super-resolution image.

Commercial implementations of these super-resolution methods have been available for more than a decade. However, they require more attention to the sample preparation as well as expert knowledge of the machinery and of the execution of the applied microscopy technique. Several new dyes have been developed to meet the experimental criteria for RESOLFT or SMLM applications. Structural illumination remains the only technique that appears more compatible with commonly used fluorophores from standard and high-resolution microscopy. However, this method usually achieves comparably significantly lower resolution gains. In general, super-resolution comes also at the cost of the so-called photon budget, and significant illumination energies are necessary. Bleaching of the fluorophores and associated phototoxicity become an obvious issue especially in live imaging. In addition, algorithms necessary to generate the higher-resolution images can introduce artefacts. These need to be identified according to the used method and mitigated mostly through changes in the algorithms and improvements in the sample preparation and imaging conditions. STED and GSD can be regarded as direct super-resolution imaging methods without the need for intensive post-processing.

The continued development of higher-resolution confocal microscopes achieves significant resolution improvement while retaining a very convenient user-friendliness. For example, the use of digital pinholes and algorithmic pixel intensity reassignment methods becomes applicable to good standard labeling methods while generating images beyond the diffraction limit. Other examples of relatively easily accessible super-resolution methods are based on the identification of fluctuations of fluorophore intensities within diffracted spots such as Super-Resolution Radial Fluctuation [127] or Super-resolution Optical Fluctuation Imaging (SOFI) [128] which reduce the amount of images necessary to around 100 rather than nearly 10,000 for standard SMLM.

The application of the fluorescent light microscopy techniques contributed to the understanding of entire viral life cycles. Fluorescence widefield and confocal microscopy techniques are now the established standards as described above for AAV. Other virus research fields provide a multitude of examples where microscopy contributes to the understanding of molecular mechanisms such as virus entry, genome integration, expression, assembly, and immunological consequences (reviewed in [129,130]). For example, TIRF microscopy was used in a combination with FRET and FRAP in temporal studies determining the interplay of host proteins with the assembly of HIV at the plasma membrane [131]. Similar analyses of spatial–temporal dynamics of interactions may be interesting for AAV capsid assembly studies. The choice of the labeling technique is key to the technical applications. Again, fluorescent proteins were fused to proteins of interest at positions retaining their functions. For example, HIV Gag or Env permit the internal insertion of FPs at positions between functional domains [132,133,134,135]. Such an internal insertion approach was used in AAV VP3 with mCherry [102]. Further in the example of HIV research, viral assembly includes the encapsulation of two copies of the genome. Labeling different sets of RNA strands with binding sites for different RNA binding proteins fused to fluorescent proteins enabled not only the counting of the number of copies included in the virus [136]. Other techniques used nanobodies for the integration steps.

Structured illumination microscopy resolution improvements are typically around a factor 2 from the diffraction limit. This makes the method suited for the analyses of particles bigger than 100 nm. SIM finds applications in the recording of single virus particles such as HIV which are around 100–120 nm in diameter [137], influenza virus [138] or vaccinia virus [139]. In AAV research, it can be used to visualize cellular compartments but less so single particles.

Super-resolution techniques enable the observation of objects in sizes from single virus particles (medium to large at the 100 nm scale) to up to single molecules in complexes. RESOLF and SMLM techniques can achieve resolutions below 50 nm, therefore allowing a more precise counting and visualization up to single molecules. For example, the molecular rearrangements of the HIV Env complexes during viral maturation were analyzed with STED and dSTORM [140,141]. Molecular structures of Herpes Simplex Virus 1 (HSV-1) have also been characterized by dSTORM [142]. STED has been combined with FCS to describe precise changes in HIV-1 Env mobility according to the maturation state [143]. Further novel labeling approaches such as the incorporation of unnatural amino acids in the target proteins (see section: labeling with click chemistry) permit the direct use of potent synthetic dyes without the need for potential spatial constrained antibodies or nanobodies [144,145]. The human parvovirus B19 replication cycle was analyzed by a proteomics study in combination with laser scanning and STED microcopy. DNA was labeled with BrdU and host cell proteins (RPA32, PCNA, RFC1, Pol α, δ, ε, ζ, η, κ, λ) with antibodies demonstrating colocalization with the cellular replication machinery (Pol α, δ, ε) and RPA32 [146]. For a more general view on super-resolution microscopy in the context of viruses, we would like to refer to recent reviews, which include among others citations for HIV, influenza, Ebola, Zika and human respiratory syncytial viruses [147,148].

The very high resolutions determining molecular rearrangements make super-resolution microscopy complementary to classical structural biology techniques. New developments include correlation microscopy studies overlaying fluorescent signals from light microscopy with structures recorded in electron microscopy such as, for example, in the visualization of single HIV particles at the virological synapse [149].

### 3.5. Super-Resolution Microscopy in the Context of AAV

As other virology research fields, AAV research began to apply super-resolution fluorescent microscopy in recent years. Next to Seisenberger et al., the group of Samulski expanded its own earlier work in 2012 and developed single-particle fluorescence imaging with 3D deconvolution and isosurface rendering using Cy5-AAV2 transducing HeLa cells. Fixing cells at 0, 2, 4, 8 and 13 h post-viral contact and staining for lysosomes with anti-Lamp1 and DNA with DAPI enabled them to spatially and temporally define the intracellular migration of AAV particles [150]. An even more detailed picture was obtained by the group of Yang, who utilized ‘single-point edge-excitation sub-diffraction’ (SPEED) [151] to study the nuclear import of AAV via the nuclear pore complex (NPC) in live cells. They used a mosaic AAV2 with the AVI peptide inserted after position 139 in VP1, BirA biotinylation and the addition of AlexFluor647-labeled streptavidin, which provided on average 36 dyes per particle while maintaining biological functionality. In the target HeLa cells, the NPC protein POM121 was GFP labeled, and dual excitation images were taken in 20 ms intervals with a spatial resolution of about 9 nm [152,153].

It should also be noted that AAV served as a key transfection reagent for super-resolution studies such as the STED imaging of in vivo mouse brains [154,155]. A further, recently introduced, optical single-particle analysis method is interferometric scattering microscopy, which is the basis for mass photometry, a method to analyze primarily the mass of purified AAV particles [156,157].

## 4. Outlook

Despite a large body of knowledge on AAV biology, fundamental questions remain to be answered. Specifically, the intricate network of protein–protein and protein–DNA interactions as well as the capsid assembly and packaging processes during AAV and rAAV genesis pose open questions regarding the underpinning mechanisms and dynamics. The transduction process is better characterized, but even here, for example, the forces and localization of the uncoating remain opaque. It is still unclear how many particles attached to the cell surface finally had their genomes expressed and what the main hurdles are. Fluorescence microscopy combining single-particle analysis in a representative number of cells is well suited to address these questions and to reveal the dynamics and subcellular and sub-nuclear localization as well as the colocalization of enabling molecules. This requires capturing multiple AAV and/or proteins of interest simultaneously at the single-particle or molecule level with respect to quantity and dynamics. Such knowledge might help to alleviate bottlenecks of rAAV production and transduction for therapeutic applications. Although AAV was among the first to be imaged at super-resolution, such data remain sparse compared to other viruses such as HIV, because the small AAV diameter and the less well understood and dense organization of the nucleus put high demands on staining and imaging technology, which were not met by user-friendly systems.

Recent technical and availability developments in super-resolution and expansion microscopy as well as the combination of electron and fluorescence microscopy will soon bring single-particle AAV microscopy into the reach of more laboratories and drive research. Specific labeling and fluorophore improvements as well as new illumination methods applied in STED microscopes permit long live-imaging studies with very high lateral resolutions around 10 nm or under (MINFLUX) [158,159], or they aim at single digit nanometer resolution (MINSTED) [160]. Compared to other live cell compatible methods, such approaches would clearly resolve single AAV particles. Furthermore, super-resolution microscopy in combination with the acquisition of further information such as fluorescence correlation spectroscopy (STED-FCS) might in the near future deliver more information on AAV–protein and AAV–particle interactions [161,162].

To gain even higher resolution, fluorescent light microscopy can be combined with electron microscopy. This technique would be classified as correlative light electron microscopy (CLEM). Fluorescent and electron microscopy are applied to the same sample. This allows for the combination of the high resolution of EM (typically up to single-digit ångström range) with the localization of labeled molecules [163]. Cryo-EM has been one of the most important microscopy methods in the elucidation of capsid structures of different AAV serotypes as well as their interaction with cellular receptors [164].

Exciting future possibilities for single AAV particle identification also come from expansion microscopy [165,166]. The principle of expansion microscopy is to expand the sample in an isotropic way typically by a factor 10 up to 100. These methods reach resolutions close to electron microscopy while preserving the possibility of standard fluorescent labeling. Localizations, interactions and structural information become accessible with standard fluorescent microscopes. An improvement in protocols permitted better expansions of single virus particles such the HSV-1 or HIV-1 to characterize the molecular distribution at their surfaces [167]. Recently, SARS-CoV-2 particles have been analyzed by a combination of expansion with super-resolution techniques such as STED and SMLM up to resolutions of 1 nm, demonstrating feasibility for small viruses such as AAV [168].

In addition, the present lower resolution techniques benefit from the development of new dyes and fluorescent proteins, which not only report location but also sense the environment or allow pulse-chase settings. Following AAV uptake and endosomal trafficking might be aided by florescent proteins suited for ratiometric pH tracking in endosomes [169]. Using convertible, switchable or activatable fluorescent proteins may aid in the determination of viral protein dynamics from nuclear protein import to assembly [120].

Collectively, techniques with single AAV particle resolution and the capacity for live cell imaging in combination with context-sensing techniques provide great potential to fill the remaining gaps in our knowledge of the AAV life cycle and recombinant AAV deployment.

## Figures and Tables

**Figure 1 viruses-15-01174-f001:**
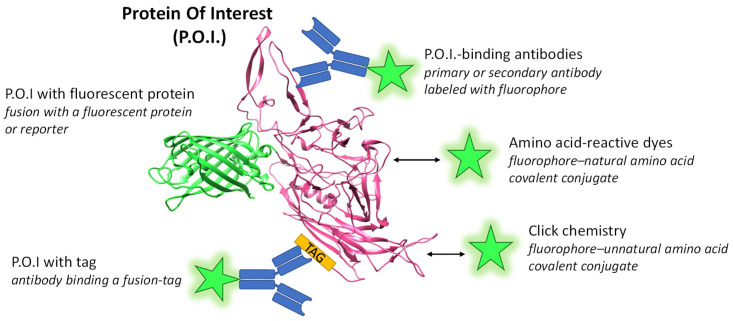
Scheme of labeling methods for a protein of interest (P.O.I.). Most commonly, proteins are covalently labeled with amine reactive derivatives of established dyes, e.g., from the Cy or AlexaFluor series. More advanced options such as coupling via click chemistry exist. A well-established specific labeling method uses antibodies that are coupled to fluorescent dyes or are recognized by a secondary fluorescently labeled antibody. Furthermore, the fusion of a gene of interest to sequences coding for a fluorescent protein leads to the in vivo expression of a labeled P.O.I. Here, a fusion protein of VP3 of AAV2 as the P.O.I. (PDB 1LP3 [4]) and GFP (PDB 6XZF) is depicted. Green stars represent fluorophores and arrows indicate a chemical coupling reaction.

**Figure 2 viruses-15-01174-f002:**
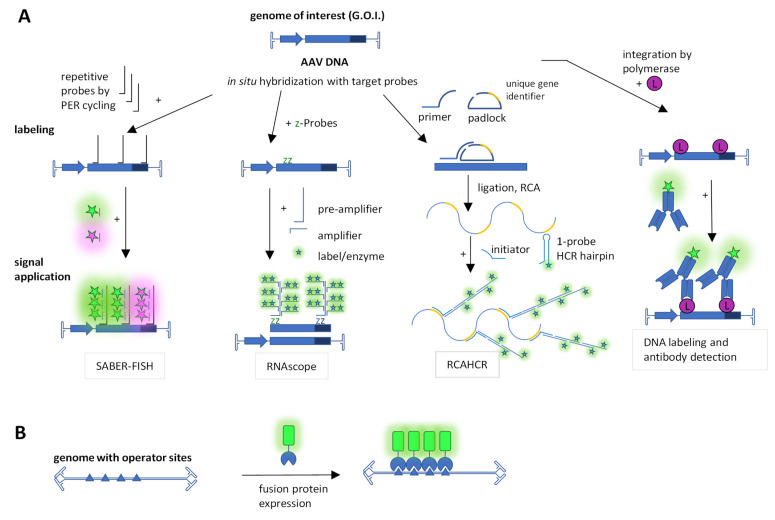
Overview of techniques for AAV DNA labeling. (**A**) In vitro labeling techniques. The SABER-FISH relies on repetitive probes generated by primer exchange reaction (PER). RNAscope uses two target probes (z-oligonucleotides) for selective detection and a tree-like assembly for signal amplification. RCAHCR uses a primer -padlock “SNAIL” combination followed by rolling circle amplification (RCA) and hybridization chain reaction (HCR). Furthermore, AAV DNA can be labeled, e.g., during in vivo synthesis with, e.g., BrdU, and this label can be then recognized by fluorescently labeled antibodies. Black or blue lines denote oligonucleotides and stars represent fluorophores, which may be of different colors e.g., green or red. (**B**) Live cell DNA imaging. Endowing DNA with multiple operator sites and expressing the respective repressor–protein fused to a fluorescent protein enabled live cell detection. Triangles represent operator sites, the partial circle a repressor and the green square a fluorescent protein. In a similar fashion, although not yet reported for AAV, CAS-GFP fusions from the CRISPR system have also been used for DNA painting based on consecutive-binding single-guide RNAs.

**Figure 3 viruses-15-01174-f003:**
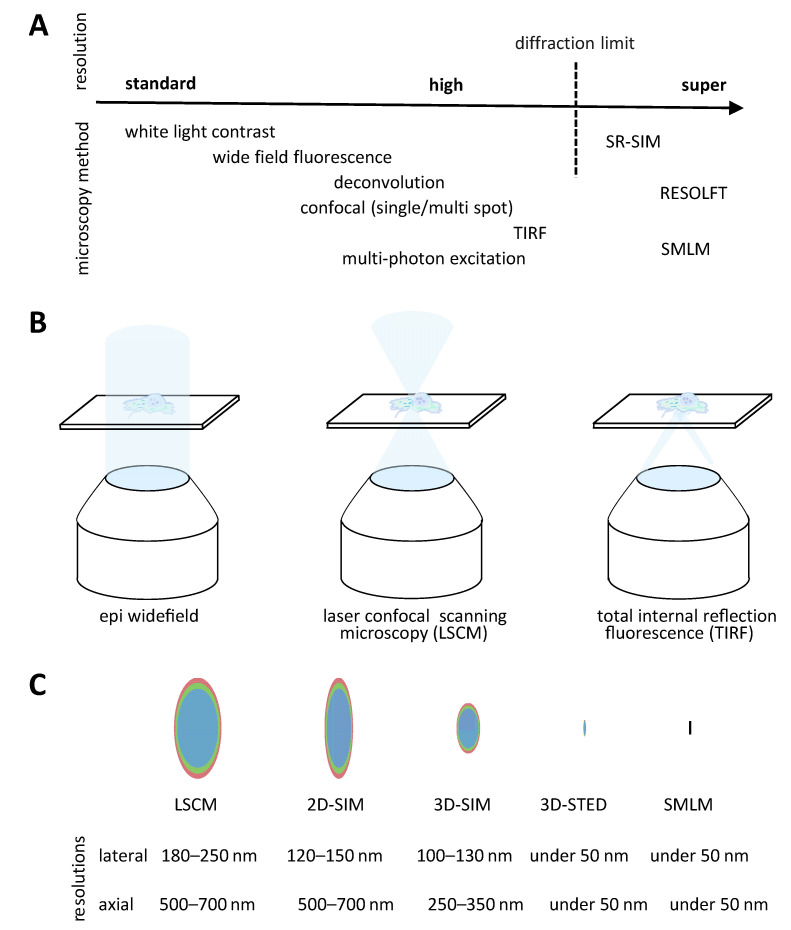
Overview of fluorescence microscopy methods. (**A**) Scheme of methods and their approximate resolution range in three typical categories. Standard and high-resolution fluorescence microscopy methods are all limited in absolute resolution by diffraction. (**B**) Scheme of light paths in standard and high resolution. High-resolution techniques typically achieve clean optical sectioning by structured illumination with or without further software deconvolution to improve contrast. These methods typically employ lenses with high magnification and numerical aperture. (**C**) Schematic representation of the resulting point-spread function of microscopy methods and the expected axial and lateral resolution range. Only super-resolution techniques achieve resolutions under the diffraction limit, typically under 100–150 nm in lateral resolution. The resolutions are dependent on the wavelength (red to blue), i.e., lower wavelengths result in higher resolutions (e.g., discussed in [116]). RESOLFT: Saturable Optical Fluorescence Transitions; SR-SIM: super-resolution structured illumination microscopy; STED: stimulated emission depletion; SMLM: single molecule localization microscopy.

**Table 1 viruses-15-01174-t001:** Overview of antibodies against AAV proteins providing the AAV serotype, the binding region and a reference. The table combines information from other reviews with additional information [69,70,71,72,73].

Virus and Protein	Antibody Name	Binding Region	Amino Acids (VP1 Numbering)	Reference
AAV1	4E4	protrusions across 2-fold axis	456–459, 492–498	[74]
AAV1	ADK1a	3-fold protrusions	448, 450, 453–457, 500	[70]
AAV1	ADK1b	2/5-fold wall; side of 3-fold	256, 258, 259, 261, 263–266, 272, 385, 386, 547, 709, 710, 716–718, 720, 722	[70]
AAV1, AAV6	5H7	center of 3-fold symmetry axis	494, 496–499, 582, 583, 588–591, 593–595, 597	[74]
AAV1, AAV3, AAV5	D3	highly exposed on VP3 surface	474–483	[69]
AAV1,2,3,5,6,7,8,9, rh10, DJ VP1, VP2, VP3	B1	buried C-terminal aa	726–734	[69]
AAV2,3	A20	2/5- fold wall and canyon	253, 254, 258, 261, 262, 264, 384, 385, 548, 556, 658–660, 708, 717	[75]
AAV2	C37-B	3-fold protrusions	492–498, 585–589	[74]
AAV2; VP1	A1	N-terminal VP1 domain	122–131	[69]
AAV2; VP1, VP2	A69	N-terminal VP2 domain	171–182	[69]
AAV5	3C5	2/5-fold wall	siteA: 254–261, 374, 375, 483, 485–492, 494, 496, 499, 500, 501 siteB: 246, 530, 532–538, 653, 654, 656, 657, 704–708	[74]
AAV5	ADK5a	2/5- fold wall	244, 246, 248–256, 263, 377, 378, 453, 456, 532, 533, 535–543, 546, 653, 654, 656, 697, 698, 704–710	[70]
AAV5	ADK5b	2/5- fold wall to 5-fold symmetry axis	248, 316–319, 443, 530–535, 540–543, 545, 546, 697, 704, 706, 708–710	[70]
AAV5	HL2476	3-fold protrusions	481, 483, 484, 576	[76]
AAV6	ADK6	3-fold protrusions and 2/5-fold wall	K531 selectivity	[77]
AAV8	ADK8	3-fold protrusions	586–591	[78]
AAV9	PAV9.1	center of 3-fold symmetry axis	496–498, 588–592	[79]
MAAP of AAV2	Anti-MAAP GAL-KKI	polyclonal	79–98 of MAAP	[11]

## Data Availability

Data sharing not applicable.

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
