# Peer review of "Fluorescence Microscopy in Adeno-Associated Virus Research"

_viruses, 2023, doi:10.3390/v15051174_

Round 1
Reviewer 1 Report
This manuscript is of exceptional merit and provides the field with an excellent resource to learn from and reference.
There are a few minor revisions, I suggest:
- On line 28, change to Parvoviridae
- Figure 1 is very helpful, however a few modifications would make it better. First - label each type of identification with a letter or the corresponding section title, additionally match the text to the section title (ex - instead of "Fluorophore - unnatural amino acid covalent conjugate" use "Click Chemistry" so that the labels on the figure more easily reference specific sections of the manuscript text). Also please provide an example fusion-tag (i.e. FLAG) shown in a separate color. Currently it appears that the antibodies depicted next to the text "antibody binding a fusion-tag" are attached to the N-terminus of the monomer.
Author Response
We thank the reviewer for the supportive comment.
We changed the wording (Parvoviridae) and adjusted figure 1 as suggested by the reviewer.
Reviewer 2 Report
This review article summarizes progress in the use of imaging approaches to visualize DNA, proteins and capsids of AAV. This is an interesting and timely topic and there have been no other reviews that cover the intersection of fields in quite the same way. The review introduces AAV, and then briefly describes the infectious cycle, referencing primary literature and other review articles. The review then introduces labeling approaches for imaging, and how they have been applied to AAV over the years. The next section deals with microscopy methods. This section is perhaps less directly relevant to AAV but includes useful information and some speculation on potential applications. Overall, the review covers the current status of the field quite well and is a comprehensive summary of what has been done. I like the fact that they have included the original articles that described trafficking and structures formed during AAV replication, and have then combined these with newer studies using more sophisticated imaging approaches. It will definitely be a useful resource for those entering the field and wanting to know what has been done. My only suggestion is that the Outlook section could be a bit more provocative about what still needs to be addressed. It would be nice if they concluded by saying what the remaining gaps are in the field and proposing which imaging approaches could be best employed to fill these gaps.
Author Response
We thank the reviewer for positive assessment of our manuscript.
We restructured the outlook and first describe, in a more provocative manner, the imaging needs and then highlight the respective developments, which could meet that demand. We considered to expanded the outlook but in the writing process we realized that a detailed problem-solution description would be lengthy and duplicate parts of the previous text. We hope that the short version fulfills the reviewers expectations.